# A Micromechanical-Based Semi-Empirical Model for Predicting the Compressive Strength Degradation of Concrete under External Sulfate Attack

**DOI:** 10.3390/ma16165542

**Published:** 2023-08-09

**Authors:** Shagang Li, Xiaotong Yu, Shanyin Yang, Hongxiang Wang, Da Chen

**Affiliations:** 1Key Laboratory of Ministry of Education for Coastal Disaster and Protection, Hohai University, Nanjing 210098, China; lishagang@hhu.edu.cn (S.L.); xiaotongyu@hhu.edu.cn (X.Y.);; 2College of Harbour, Coastal and Offshore Engineering, Hohai University, Nanjing 210098, China; 3Yangtze Institute for Conservation and Development, Hohai University, Nanjing 210098, China

**Keywords:** concrete, external sulfate attack, compressive strength, degradation, model

## Abstract

As one of the most harmful ions in the environment, sulfate could cause the deformation and material deterioration of concrete structures. Models that accurately describe the whole chemo–transport–mechanical process of an external sulfate attack (ESA) require substantial computational work and contain complex parameters. This paper proposes a semi-empirical model based on micromechanical theory for predicting the compressive strength degradation of concrete under an ESA with basic properties of the undamaged material and limited computational effort. A simplified exponential function is developed for the total amount of the invading sulfate, and a second-order equation governs the chemical reaction. A micromechanical model is implemented to solve the mechanical response caused by an ESA. The model is able to describe the compressive stress–strain behavior of concrete subject to uniaxial loading in good agreement with the experimental results. For the case of a sulfate-attacked material, the relationship between compressive strength and expansion is calculated and validated by the test results. Finally, the deterioration process of compressive strength is predicted with the test results of deformation.

## 1. Introduction

Massive concrete structures, such as piers, breakwaters, artificial islands, bridges, and wind turbine foundations, have been widely applied in marine and salt-lake environments. Long service life is required owing to the significant social and economic benefits of these structures. However, some potential degradation mechanisms in the aggressive environment may lead to the premature failure of these structures [1,2]. Among them, an external sulfate attack (ESA) could directly damage concrete. Sulfate ions diffuse into the concrete pore system and then react with the cement hydrate products. The main insoluble reaction product, ettringite, which has a much larger volume than solid reactants, fills pores and leads to expansion and cracking. If the structure is cracked, seawater will directly contact the interior material, causing severer concrete damage and even reinforcement corrosion. Therefore, it is of great importance to assess the potential damage to concrete under an ESA so that engineering structures can be designed and repaired to ensure the required service life after long-term degradation.

Compressive strength and its evolution are the most important parameters considered in the design and analysis of concrete structures [3,4]. The effect of an ESA on compressive strength has been reported in many experimental investigations [5,6,7]. The compressive strength evolution of concrete with different cement types and water–cement ratios was evaluated by some authors [8,9]. Concrete constructed with a higher cement C_3_A content and water–cement ratio usually experienced a more considerable reduction of compressive strength. Environmental conditions, such as sulfate concentration and temperature, were also proven to affect the deterioration process of an ESA significantly [10,11]. A sharp strength reduction was observed after long-term exposure to a high-concentration solution. Based on the test results, some empirical models were proposed to predict the residual compressive strength after a long-term ESA. Yu et al. [12] used a binomial function to describe the relationships between compressive strength and corrosion age and fitted this formula with data from other studies. Cheng et al. [13] divided the attacked concrete into the damaged and healthy areas and established the relationship between the compressive strength of the damaged area and the integral area of sulfate ions distributions. However, extensive destructive tests, such as compressive strength and sulfate content tests, are required to calibrate the parameters of these models. In addition, the value of these parameters is usually limited to specific materials and external environments and may need an update in a new working condition.

Models considering the complex process of ion diffusion, chemical reaction, expansion, and damage during an ESA have also been widely researched. Usually, a modified Fick’s law was used to describe the diffusion process, and the reaction was assumed to be one or two orders with a global sulfate phase–aluminate phase form [14]. There are two different assumptions about the expansive process. Some considered that the expansion was caused by the volume variation between the reactants and the reaction products [15]. In contrast, others assumed that it was the result of the crystallization pressure of ettringite [16]. The cracking damage induced by an ESA was often described with a parameter related to the expansive strain [17]. Some studies also modeled the effects of pore filling and cracking damage on the diffusion coefficient [18]. Recently, some models [19] coupled the process of calcium leaching and an ESA and simulated their interactions. These models could describe the deleterious process of an ESA in detail but might not be suitable for engineering applications. A series of partial differential equations, ordinary differential equations, and algebraic equations in these models are difficult to solve in closed form. The numerical solution of these equations means a substantial computational cost. Furthermore, there is no consensus on the value of some essential model parameters, such as diffusion coefficient, reaction coefficient, etc. [20,21].

This paper aims to develop a semi-empirical model to predict the degradation of compressive strength under an ESA with limited parameters and computational effort. For this purpose, a simplified exponential function and a second-order equation are used for the diffusion and reaction processes. The volume variation theory is adopted to describe the expansion caused by an ESA. The mechanical response and damage due to an ESA and external load is computed based on micromechanical theory. The uniaxial compressive process of a material without an ESA is first simulated and compared with the experiments. Later, the damage process of compressive strength due to an ESA is simulated and validated, and the compressive strength at different corrosion ages is predicted based on the results of deformation tests.

## 2. Description of the Simplified Model

### 2.1. Simplified Microstructure of Concrete

A microstructure analysis of concrete under an ESA showed that microcracks first appeared on the surface directly contacted with the corrosion solution due to the chemical reactions between sulfate ions and pore solution [11]. After, cracks propagated into the interior of the material with sulfate ion diffusion (Figure 1). The accumulation of sulfate ions directly leads to cracking, and the propagation of cracks makes the diffusion properties of the material change over time and space. As a result, the simulation of the detailed ESA process becomes complicated and time-consuming due to the interactions between ion diffusion and crack propagation. Then, simplification of the diffusion and damage process is necessary.

In the domain of engineering design, it is a common practice to employ the average value of material properties throughout the entire zone, as determining the spatial distribution of such properties can be a challenging task. Therefore, it is assumed that the invading sulfate ions are uniformly distributed in the corroded concrete, and microcracks are scattered into the whole material. Then, concrete is considered a composite composed of undamaged solid matrix and cracks, as illustrated in Figure 2.

In a representative volume element (RVE) of the material, the total volume *Ω* is the superposition of matrix volume *Ω_m_* and crack volume *Ω_c_*. Then, their volumetric fractions *f_m_* and *f_c_* are defined as:(1)$$f_{m}=\frac{\Omega_{m}}{\Omega };f_{c}=\frac{\Omega_{c}}{\Omega }$$

Since medium with orthogonal cracks and randomly oriented cracks is confirmed to have similar properties [22], three orthogonal families of cracks are assumed, with each family of cracks having the same normal. In addition, the volume fraction of the three families *f_ci_* (*i* = 1, 2, 3) is considered to be the same:(2)$$f_{c1}=f_{c2}=f_{c3}=\frac{f_{c}}{3}$$

In the micromechanical modeling of concrete [23,24], cracks are usually assumed to be penny-shaped with radius *a_i_* and crack opening *c_i_*. The volume fraction *f_ci_* and aspect ratio *X_i_* of the *i*-th family, which describe the development of the microstructure, are defined as [25]:(3)$$f_{ci}=\frac{4}{3}\pi c_{i}n_{i}a_{i}^{2}$$
(4)$$X_{i}=\frac{c_{i}}{a_{i}}$$
where *n_i_* is the crack density of the *i*-th family.

### 2.2. Diffusion of Sulfate Ions

When the material is fully immersed in sulfate solution, the sulfate ions are transported in the form of diffusion. It is assumed that the temporal and spatial distribution of sulfate ions can be described by Fick’s second law and mass conservation law [14]:(5)$$\frac{𝜕c_{s}}{𝜕t}=\nabla \cdot \left(D_{s}\nabla c_{s}\right)+c_{sr}$$
where *c_s_* is the sulfate concentration at time *t*, *D_s_* is the diffusion coefficient of sulfate, *c_sr_* is the reacted sulfate concentration.

When the reaction is ignored and the diffusion coefficient *D_s_* is considered constant, the total amount of invading sulfate ions can be solved analytically [26]:(6)$$\frac{M_{t}}{M_{\infty }}=\{\begin{array}{c} 1-\overset{\infty }{\underset{n=1}{\sum }}\frac{4}{J_{n}^{2}}\exp\left(-D_{s}J_{n}^{2}t/R^{2}\right),\;\mathrm{diffusion in a long circular cylinder}\\ 1-\frac{6}{\pi^{2}}\overset{\infty }{\underset{n=1}{\sum }}\frac{1}{n^{2}}\exp\left(-D_{s}n^{2}\pi^{2}t/R^{2}\right),\;\mathrm{diffusion in a sphere} \end{array}$$
where *M_t_* is the total quantity of sulfate entering the material at time *t*, $M_{\infty }$ is the corresponding amount after infinite time, *J_n_* are the roots of the Bessel function of the first kind of order zero, *R* is the size of the material.

Referring to Equation (6), a simplified exponential function is proposed to empirically describe the total amount of sulfate invading the material at different time:(7)$$\frac{M_{t}}{M_{\infty }}=1-\exp\left(-\alpha t\right)$$
where *α* is a parameter related to the ion diffusion property of the material.

Figure 3 compares the results calculated from Equations (6) and (7) with different diffusion coefficient *D_s_* and material size *a*. Although the results are not the same, Equation (7) successfully reflects the growth law of sulfate concentration with time.

Then, the average sulfate concentration of the material $\bar{c_{s}}$ at time *t* can be obtained from Equation (7):(8)$$\bar{c_{s}}=\bar{c_{\infty }}\left[1-\exp\left(-\alpha t\right)\right]$$
where $\bar{c_{\infty }}$ is the average sulfate concentration after infinite time.

### 2.3. Chemical Reaction

The expansion caused by an ESA is mainly due to the reactions between sulfate and different aluminate phases, which can be merged in a global aluminate phase-sulfate phase form [27]:(9)$$\mathrm{CA}+q\bar{S}\to C_{6}{\mathrm{AS}}_{3}H_{32}$$
where CA and $\bar{S}$ represent, respectively, the equivalent aluminate phase and sulfate phase; *q* is the weighted average stoichiometric coefficient of the reaction. It is observed that the reaction product ettringite mainly precipitates in the capillary pores [28]. As a result, it is assumed that the chemical reaction occurs in the capillary pores from which cracks also originate.

To describe the reaction between sulfate and aluminate, a second-order reaction equation, which assumes that the reaction rate is proportional to reactant concentrations, is employed [15]:(10)$$\frac{dc_{e}}{dt}=kc_{al}c_{s}$$
where *k* is the chemical reaction rate constant and *c_e_* and *c_al_* are the concentrations of ettringite and equivalent aluminate phase, respectively.

The consumption rate of sulfate and aluminate is written as follows:(11)$$\frac{dc_{s}}{dt}=-\frac{k}{q}c_{al}c_{s}$$
(12)$$\frac{dc_{al}}{dt}=-kc_{al}c_{s}$$

Inserting Equation (8) into Equation (12) yields:(13)$$\frac{dc_{al}}{dt}=-k\bar{c_{\infty }}c_{al}\left[1-\exp\left(-\alpha t\right)\right]$$

With the initial condition $c_{al}{/}_{t=0}=c_{al0}$, the following equation is obtained:(14)$$c_{al}=c_{al0}\exp\left\{-k\bar{c_{\infty }}\left[\left(t+\exp\left(-\alpha t\right)/\alpha -1/\alpha \right)\right]\right\}$$

By substituting Equations (8) and (14) into Equation (10), Equation (10) can be rewritten as:(15)$$\frac{dc_{e}}{dt}=k\bar{c_{\infty }}c_{al0}\left[1-\exp\left(-\alpha t\right)\right]\exp\left\{-k\bar{c_{\infty }}\left[\left(t+\exp\left(-\alpha t\right)/\alpha -1/\alpha \right)\right]\right\}$$

Assuming that no ettringite exists before an ESA ($c_{e}{/}_{t=0}=c_{e0}$), the concentration of ettringite can be deduced by integrating Equation (15):(16)$$c_{e}=c_{al0}-c_{al0}\exp\left\{-k\bar{c_{\infty }}\left[\left(t+\exp\left(-\alpha t\right)/\alpha -1/\alpha \right)\right]\right\}$$

### 2.4. Expansion

According to the reaction described in Equation (9), the volumetric change due to the difference in molar volume can be written as [27]:(17)$${\left(\frac{\Delta V}{V}\right)}_{r}=\frac{V_{e}}{V_{al}+qV_{s}}-1$$
where *V_e_*, *V_al_*, and *V_s_* are the molar volume of ettringite, aluminate phase, and sulfate phase, respectively.

Based on the volume averaging theory [29], the volumetric change of solid phase caused by the chemical reaction in the whole material is calculated with the amount of the reaction product:(18)$$\varepsilon_{V}^{0}={\left(\frac{\Delta V}{V}\right)}_{r}c_{e}$$

The solid product ettringite will precipitate in the capillary pores. Since ettringite is needle-shaped and generates stress after touching the pore wall, expansion usually begins before the pores are fully filled [30]. Hence, only a fraction of the pore volume *β* is considered to be precipitated by the reaction product, and the total volumetric strain can be represented as:(19)$$\varepsilon_{V}=\max\left\{\left(\varepsilon_{V}^{0}-\beta f_{c}\right),0\right\}$$

In full immersion case, the internal pressure *P* is mainly caused by chemical sulfate attack, which can be expressed as [31]:(20)$$P=K_{e}\varepsilon_{V}$$
where *K_e_* is the bulk modulus of ettringite.

When $\varepsilon_{V}^{0}⩾\beta f_{c}$, combining Equations (16)–(20) yields:(21)$$\frac{P}{K_{e}}+\beta f_{c}={\left(\frac{\Delta V}{V}\right)}_{r}c_{al0}-{\left(\frac{\Delta V}{V}\right)}_{r}c_{al0}\exp\left\{-k\bar{c_{\infty }}\left[\left(t+\exp\left(-\alpha t\right)/\alpha -1/\alpha \right)\right]\right\}$$

### 2.5. Constitutive Equation and Damage Criterion

The material under an ESA may suffer from both external load and chemical corrosion. The relationship between macroscopic stress **Σ** and strain **E** is written as [25]:(22)Σ=ℂhom:E−BP
where ℂhom is the effective drained stiffness tensor of the RVE and **B** is the Biot tensor, which describes the porosity property of the RVE. The effective drained stiffness tensor ℂhom can be calculated according to the Mori–Tanaka scheme [32]:(23)ℂhom=fmℂm:[fmI+∑i=13fci(I−Sci)−1]−1
where ℂm is the stiffness of the solid matrix, I is the fourth-order symmetric unit tensor, and Sci is the Eshelby tensor of the *i*-th crack family.

The Biot tensor **B** is defined as follows [33]:(24)B=I−ℂm−1:ℂhom:I
where **I** is the second-order unit tensor.

The damage criterion is formulated based on thermodynamic and linear fracture mechanics theory [25]:(25)$$G_{\epsilon }^{i}-G_{c}^{i}⩽0;{\dot{\epsilon }}_{i}⩾0;\left(G_{\epsilon }^{i}-G_{c}^{i}\right){\dot{\epsilon }}_{i}=0$$
where $\epsilon_{i}$ is the crack density parameter, which identifies the damage state of the cracked porous media:(26)$$\epsilon_{i}=n_{i}a_{i}^{3}$$

The damage process is described by crack propagation without considering the generation, opening, and closing of cracks. Therefore, the crack opening *c_i_* and crack density *n_i_* are constant, and the crack density parameter $\epsilon_{i}$ is directly related to the crack radius *a_i_*. It should be noted that this model is only suitable when ordinary Portland cement is the primary binding material. When large amounts of supplementary cementitious materials, such as fly ash, slag, and silica fume, are used in concrete, significant variations of material properties will be caused by cement hydration during the process of an ESA.

The energy release rate Gϵi represents the driving force of the damage process:(27)Gϵi=𝜕Ψ𝜕ϵi=−12〈E+Pℂm−1:I〉:𝜕ℂhom𝜕ϵi:〈E+Pℂm−1:I〉
where 〈⋅〉 is the Macaulay brackets.

The critical energy release rate $G_{c}^{i}$ is related to the loading history, crack size, observation size, and material properties. Ignoring the interaction effects between cracks, $G_{c}^{i}$ is a function of $\epsilon_{i}$:(28)$$G_{c}^{i}=\frac{2\pi }{3}G_{f}{\left(\frac{n_{i}}{\epsilon_{i}}\right)}^{1/3}$$

## 3. The Uniaxial Compressive Process Simulation of Concrete without Chemical Damage

### 3.1. Uniaxial Compressive Process Simulation

In the case of the uniaxial compressive test of concrete without chemical damage, the internal pressure *P* is equal to 0. If the loading process is controlled by strain, the macroscopic strain **E** equals the applied value **E***_app_*. The macroscopic stress **Σ***_app_* results from the applied strain **E***_app_* can be calculated from Equation (22):(29)Σapp=ℂhom:Eapp

Considering that the applied load is perpendicular to the first family of cracks, only the remaining two families of cracks grow during the loading process. Since the initial size and shape of the cracks are identical, the two families have the same propagation during the loading process. As a result, the crack pattern of the uniaxial compressive simulation is similar to the splitting cracks observed in the tests. Combining Equations (27)–(29) yields the relationship between external load and crack density parameter:(30)−12〈Eapp〉:𝜕ℂhom𝜕ϵi:〈Eapp〉⩽2π3Gf(niϵi)1/3(i=2,3)

### 3.2. Model Parameter Calculation

In general, the input parameters of this model can be determined by basic material properties and existing equations in papers. In this work, the elastic modulus and compressive strength of undamaged material *Y_in_* and *σ_c_*_,*in*_ are obtained from the test results. The compressive strain *ε_c_*_,*in*_ corresponding to the compressive strength *σ_c_*_,*in*_ in a uniaxial compressive test is calculated according to the empirical formulas in the literature [34,35]:(31)$$\varepsilon_{c,in}=\{\begin{array}{c} 80\sigma_{c,in}\times 10^{-6},\mathrm{for mortar}\\ \left(0.0546+0.003717\sigma_{c,in}\right)\times 10^{-2},\mathrm{for concrete} \end{array}$$

The undamaged material is assumed to be an isotropic material. Consequently, three crack families have the same initial volume fraction $f_{c1,in}=f_{c2,in}=f_{c3,in}=\frac{1}{3}f_{c,in}$ and aspect ratio $X_{1,in}=X_{2,in}=X_{3,in}=X_{in}$. The crack thickness *c_i_* is taken as 0.1 μm, which is on the same order of magnitude as the diameter of capillary porosity [36]. Since less than half of the capillary pore volume is observed to be filled by ettringite during the corrosion process [28], the initial crack volume fraction *f_c_*_,*in*_ is set as 45% of the capillary porosity estimated with Power’s law:(32)$$f_{c,in}=0.45max\left(f_{cp}\frac{w/c-0.39\alpha }{w/c+0.32},0\right)\times 100\%$$
where w/c is the water-cement ratio, *f_cp_* is the volume fraction of the cement paste in the material. The degree of hydration *α* is determined from w/c based on the test results of Chen and Wu [37].

Other parameters, such as the elastic modulus *Y_m_* of the solid matrix, the initial crack aspect ratio *X_in_*, and the fracture energy *G_f_*, are determined from the model with the input parameters. The elastic modulus *Y_m_* of the solid matrix is obtained with Equation (23). By imposing that the initial compressive strength and the corresponding compressive strain of the material *σ_c_*_,*in*_ and *ε_c_*_,*in*_ are equal to the ones of the undamaged material, the initial crack aspect ratio *X_in_* and the fracture energy *G_f_* are calculated.

### 3.3. Comparison with Uniaxial Compressive Tests

Four independent uniaxial compressive tests of cement-based materials without chemical reactions reported in the literature are simulated to verify the uniaxial compressive part of the model. Kohees et al. [34] cast cylindrical mortar samples with 38 mm diameter and 76 mm height. Geelong cement and a w/c ratio of 0.6 were used. The specimens were tested with a 1000 kN universal testing machine with a strain rate of 0.2 mm/min. Yu et al. [38] constructed the mortar samples utilizing PO 42.5 cement with a w/c ratio of 0.5. The specimens were cast into Φ 50 mm × 100 mm cylinders. The tests were conducted on a RE-8060 electro-hydraulic servo universal testing machine with a strain rate of 0.001 mm/s. Yi et al. [39] prepared the concrete samples using ASTM Type I cement; Φ 100 mm × 200 mm cylinders were cast with a w/c ratio of 0.54. The axial compressive load was applied using a 2500 kN universal testing machine with a strain rate of 0.003 mm/s. Tasnimi [40] prepared Φ 150 mm × 300 mm cylindrical concrete samples. Specimens were tested in a 3000 kN closed-loop servo-hydraulic compressive testing machine with a load rate of 4.30 kN/s. Table 1 lists the model parameters obtained from these experiments.

In this work, codes written in MATLAB were used to solve the model with parameters listed in Table 1. The comparisons of the uniaxial compressive stress–strain relationship between the proposed model and experimental data are shown in Figure 4. The simulation results of mortar and concrete reflect the non-linear behavior under uniaxial compression where both the initial ascending and the subsequent descending branches are presented. Moreover, the difference between the simulated and tested value of compressive strength and elastic modulus is less than 10%. Of course, the simulation curve is not completely consistent with the test curve. It is inevitable since the stress–strain curves are affected by many factors, such as material properties, size of specimens, load method, etc. [39]. It is difficult to include all these factors in a simple model. Generally speaking, the proposed model could approximate the uniaxial stress-strain relationship of concrete with limited input parameters.

## 4. Compressive Strength Prediction of Concrete under ESA

### 4.1. Experimental Procedure

Ordinary Portland cement (P.O.42.5) with 7.27 wt% C_3_A produced by Anhui Conch Company was used in this study. The chemical composition of the cement is given in Table 2. River sand with the fine modulus of 2.63 was used as the aggregate. The water to cement ratio and sand to cement ratio were set as 0.55 and 3, respectively.

Mortar specimens with dimensions 70.7 mm × 70.7 mm × 70.7 mm were molded. To ensure that the deformation and strength degradation were caused by ESA at an equivalent degree, expansion and compressive strength were tested with the same sample size. After demolding, four surfaces of the cubes were sealed with paraffin, and only two opposite faces were in contact with the sulfate solution for one-dimensional diffusion. Later, the specimens were cured in saturated lime water for 56 d. The casting and curing procedure were operated according to the Test Code for Hydraulic concrete (SL/T, 352) [41]. When initial curing was finished, the specimens were divided into two groups, one immersed in pure water and the other immersed in 5% Na_2_SO_4_ solution. The immersion solutions were renewed every month until 360 d.

For expansion measurements, two brass studs were glued on the opposite erosion surfaces of three specimens from each group after demolding. The distance between the two brass studs was measured by a Vernier caliper with an accuracy of 0.01 mm every month. The expansion of the specimens *E_t_* was calculated as follows:(33)$$E_{t}=\frac{L_{t}-L_{0}}{L_{0}-2\Delta }$$
where *L*_0_ and *L_t_* is, respectively, the distance between the two brass studs before immersed in the solutions and immersed for time *t*; Δ is the length of the brass studs.

For compressive strength measurements, specimens were taken out of the solutions every month and the paraffin on the surface was removed. Later, the surface of the specimens was cleaned and kept moist. Uniaxial compressive tests were performed with an SHT-4305 electro-hydraulic servo universal testing machine to determine the strength, which was repeated for three specimens. Force control at the loading rate of 2 kN/s was adopted in the tests. The compressive strength of the specimens *σ_c_* was calculated as follows:(34)$$\sigma_{c}=\frac{P}{A}$$
where *P* was the failure load; *A* was the compression area of the specimens.

The expansion of specimens immersed in pure water and sulfate solution is presented in Figure 5a. The mortar exposed to water slightly expanded during the one-year immersion. Since the pores of the sample are always filled with water during the hydration process, drying shrinkage was prevented. In addition, the glue might swell when absorbing water. Therefore, the expansion in the water needed to be subtracted when analyzing the actual expansion during ESA. The specimens immersed in sulfate solution showed more significant expansion than in water. The expansion before 90 d was relatively small. Later, rapid growth appeared, and the growth rate increased over time. According to the former study, the ample precipitation of AFt was frequently cited as an important source of the expansion [42]. Furthermore, the expansion was also attributed to the cracks that generated on the ends of the samples around the studs [43].

Figure 5b illustrates the evolution of compressive strength of mortar specimens subjected to water and sulfate solution for 360 d. In pure water, the compressive strength increased before 150 d and then fluctuated within a specific range at the later stage. The enhancement of the compressive strength at an early age was due to further cement hydration, which densified the microstructure of the cement paste [12]. The compressive strength of samples in sulfate solution also increased before 120 d and was larger than in pure water at this stage. Besides cement hydration, the ettringite and gypsum produced by the reaction between sulfate and cement hydration products further compacted the microstructure and enhanced the strength [44]. Subsequently, severe deterioration was caused by ESA, and the compressive strength decreased with the immersion time.

### 4.2. The Sulfate Degradation Process Simulation

The damage process of ESA was simulated by considering that the reaction product ettringite filled the pores and microcracks of the material and caused an internal pressure *P*. The material was free of stress (∑=0) in the case without external load, and the macroscopic strain **E***^P^* was determined from Equations (22) and (24):(35)EP=PI:(ℂhom−1−ℂm−1)

Since the three families of cracks had the same initial state, they were considered to have the same propagation under symmetric loading conditions. Inserting Equation (35) into Equation (25) yielded:(36)−12〈PI:ℂhom−1〉:𝜕ℂhom𝜕ϵi:〈PI:ℂhom−1〉⩽2π3Gf(niϵi)1/3(i=1,2,3)

In addition to the global swelling of the entire material, the length change of the specimens was also affected by cracks [43]. As a result, the contribution of the crack volume increase to the expansion of the material was also implemented. Since the fact that small cracks can converge into large cracks was ignored in this model, the volume fraction of cracks *f_c_* was overestimated under the same damage. Here, a reduction factor *γ* was considered when calculating the expansion *E_t_*:(37)$$E_{t}=\frac{1}{3}\left[\mathrm{tr}\left(E^{P}\right)+\gamma \left(f_{c}-f_{c,in}\right)\right]$$

The damage process of ESA could be simulated with this model: in the initial stage, the invading sulfate ions reacted with the material to form ettringite and deposit in the initial cracks (Equation (16)). The volume of ettringite increased with sulfate amount. When the volume variation due to the sulfate reaction exceeded the allowed volume of the cracks *βf_c_*, pressure on the solid matrix *P* was generated (Equation (21)), and the material began to expand (Equation (35)). However, the crack radius *a_i_* remained the initial value since the energy release rate $G_{\varepsilon }^{i}$ was still within the threshold (Equation (36)).

As sulfate erosion intensified, the internal pressure *P* reached the allowable value calculated from Equation (36) with the energy release rate Gεi equaling the critical energy release rate Gci. Afterward, the damage process of the material began, accompanied by an increase in crack radius *a_i_* and a decrease in stiffness ℂhom (Equation (23)). As a result, the maximum internal pressure that the material could withstand decreases, and the growth rate of expansion *E_t_* increased (Figure 6).

To study the effect of ESA on compressive strength, uniaxial compressive tests of corroded concrete have been simulated by repeating the process of Section 3.1. The crack radius *a_i_* and internal pressure *P* calculated in the free expansion case were used as the corresponding initial value. Before further crack propagation, the macroscopic strain caused by chemical damage **E***^P^* was first balanced by the applied strain. Therefore, the initial strain shown in Figure 7 increased with the initial crack radius. In addition, the compressive strength *σ_c_* decreased due to the damage caused by the chemical reaction.

### 4.3. The Case Study

In addition to the tests in this study, five independent experiments of ESA reported in the literature were simulated here to verify the proposed model. Sodium sulfate was used as the corrosion medium, and full immersion was adopted as the corrosion method by all the authors. However, the concentration of the corrosion solution and the immersion period differed from one test to another. A brief description of the experimental information is shown in Table 3.

With the same method in Section 3.2, the model parameters can be obtained from the material composition, the elastic modulus, and the compressive strength of undamaged material. These parameters are listed in Table 4.

The simulation results of the degradation curve of compressive strength with expansion are shown in Figure 8. Here, the compressive strength loss meant the reduction of compressive strength compared to undamaged material. Possible strength increase in the initial stage was ignored in the model since more attention was paid to the damage caused by ESA. The experimental results are also shown in dots with the same color for comparison. It can be seen that the expansion increased with the decrease in compressive strength for both mortar and concrete, which might be attributed to the fact that the degradation of strength led to the loss of resistance against expansive crystallization [48]. Generally speaking, the simulation data reflected the trend of experimental results well.

### 4.4. Prediction of Compressive Strength Deterioration

As illustrated above, the relationship between crack radius, expansion, compressive strength, and pressure can be analyzed based on the proposed model. Compared to compressive strength, the expansion due to ESA is easier to monitor for in situ concrete structures. Therefore, compressive strength deterioration was predicted in this part with the test results on the length change of the samples.

For the convenience of fitting, Equation (21) was simplified as:(38)$$\frac{P}{K_{e}}+\beta f_{c}=A-A\exp\left\{-B\left[t+C\exp\left(-t/C\right)-C\right]\right\}$$
where $A={\left(\frac{\Delta V}{V}\right)}_{r}c_{al0}$, $B=k\bar{c_{\infty }}$, $C=\frac{1}{\alpha }$.

Parameters *A*, *B,* and *C* were fitted from the test results on the expansion of the material exposed to ESA with Equation (38). Table 5 lists the fitting results. It was clear that the three model parameters *A*, *B*, and *C* varied evidently, which was caused by the difference in the cement type, material composition, sample size, and concentration of the corrosion solution used by the researchers.

Figure 9 presents the numerical simulations of compressive strength loss at different exposure ages with the parameters listed in Table 5. In the initial stage, there was insufficient expansive pressure to destroy the material because the accumulated sulfate ion concentration was still deficient. Later, more and more sulfate ions entered the material, causing expansion and cracking, thus reducing the strength. It should be noted that the deterioration law of compressive strength in different papers was not the same with different experimental designs. The numerical results demonstrated a good agreement with the test results of previous research conducted by Yu et al., Zhao et al., and Xie et al. Nonetheless, the simulated strength degradation at the later stages of erosion was smaller than the test results in this study. This disparity can be attributed to the assumption in the damage criterion that there was no interaction effect between cracks.

## 5. Conclusions

In this study, a semi-empirical model has been established to predict the compressive strength degradation of concrete subject to an ESA with basic properties of the undamaged material and limited computational effort. The total amount of sulfate in the pore solution is simplified as an exponential function. A second-order equation is also implemented to describe the chemical reaction. The damage caused by the sulfate reaction is represented by the propagation of the cracks. Numerical simulations are compared to the experimental data of both mortar and concrete from the literature.

Using only basic properties of the undamaged material and existing equations from the literature, the proposed model simulated the uniaxial compressive stress–strain relationship of undamaged material well. The model can detail the non-linear behavior under uniaxial compression, presenting both the initial ascending and subsequent descending branches.

The degradation of compressive strength with expansion due to an ESA was also investigated. The calculated results agree well with the test results. In addition, the prediction of compressive strength at different corrosion ages was achieved by fitting the parameters of the exponential function with the test results on length change. The proposed model has the potential to monitor the compressive strength variation of in situ concrete structures under ESA and issue timely warnings for repair when the strength degradation surpasses a pre-determined threshold. However, the model has certain limitations, which call for future research to focus on developing models for cement-based materials subjected to coupling of ESA with other durability problems.

## Figures and Tables

**Figure 1 materials-16-05542-f001:**
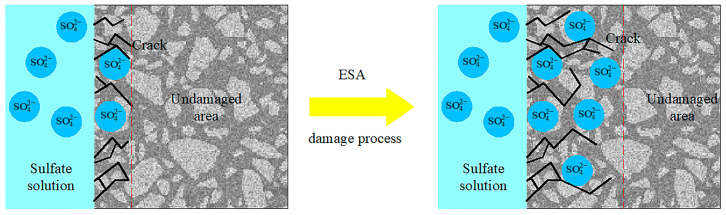
Damage process of concrete under ESA.

**Figure 2 materials-16-05542-f002:**
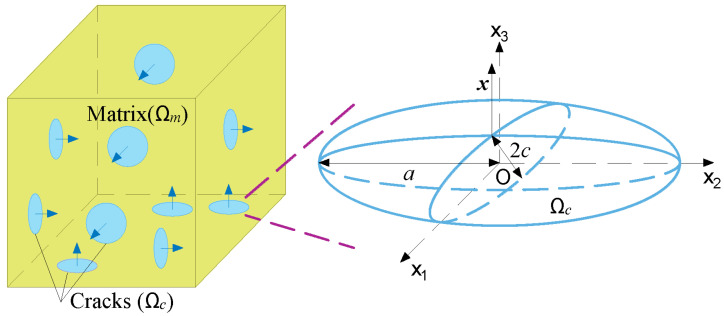
Simplified microstructure of concrete.

**Figure 3 materials-16-05542-f003:**
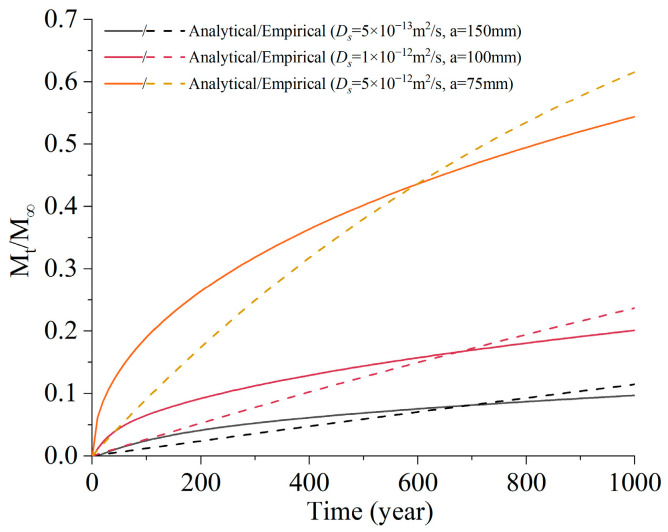
Comparison between the results of the analytical and simplified equation.

**Figure 4 materials-16-05542-f004:**
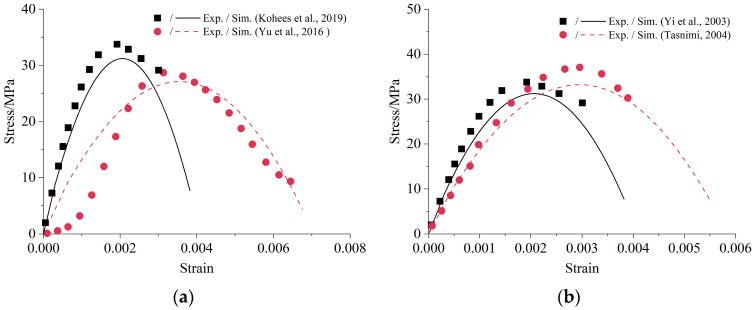
Comparison of the uniaxial compressive stress-strain relationship: (**a**) Mortar [34,38]; (**b**) Concrete [39,40].

**Figure 5 materials-16-05542-f005:**
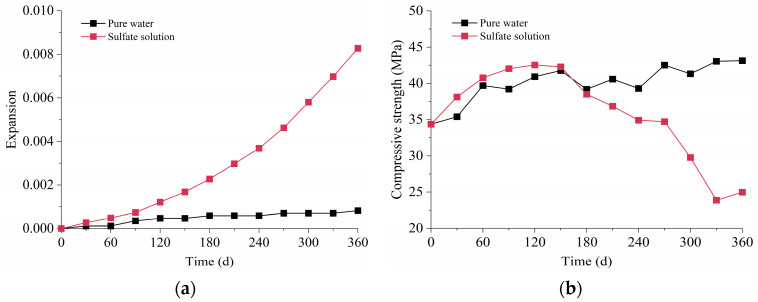
Test results of mortar immersed in pure water and sulfate solution: (**a**) Expansion; (**b**) Compressive strength.

**Figure 6 materials-16-05542-f006:**
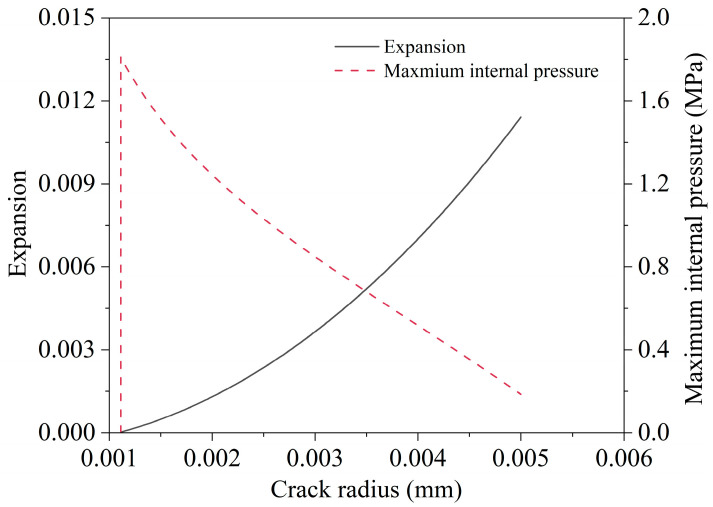
Relationship between expansion, internal pressure, and crack radius.

**Figure 7 materials-16-05542-f007:**
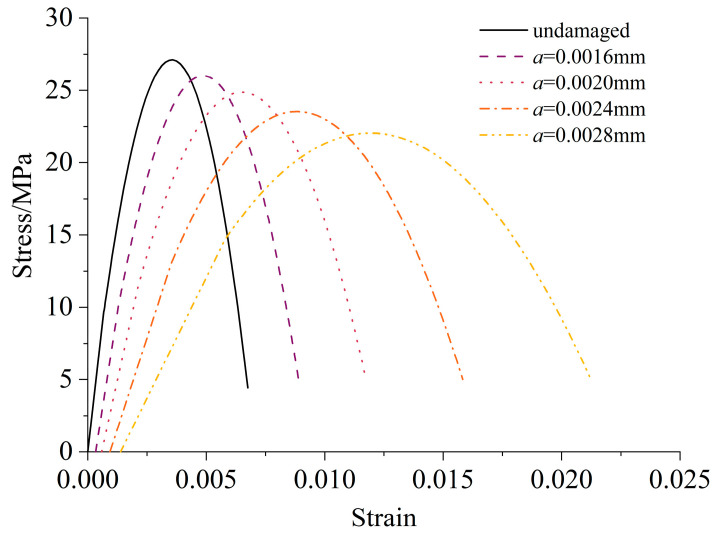
Compressive stress–strain relationship under ESA.

**Figure 8 materials-16-05542-f008:**
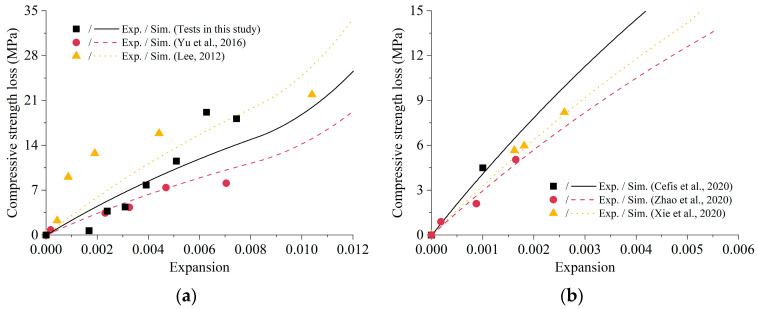
Comparison of compressive strength loss–expansion relationship under ESA: (**a**) Mortar [38,45]; (**b**) Concrete [44,46,47].

**Figure 9 materials-16-05542-f009:**
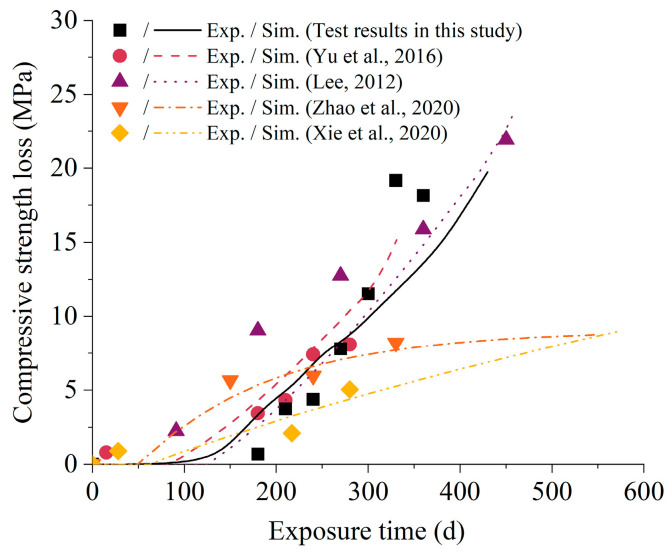
Prediction of compressive strength loss at different exposure time [38,44,45,47].

**Table 1 materials-16-05542-t001:** Model parameters for uniaxial compressive test simulation.

	Kohees et al. [34]	Yu et al. [38]	Yi et al. [39]	Tasnimi [40]
Material	Mortar	Mortar	Concrete	Concrete
*Y_in_* (GPa)	23.21	14.40	28.88	21.42
*Y_m_* (GPa)	31.42	17.85	36.32	25.67
*σ_c_*_,*in*_ (MPa)	38.00	29.18	33.78	37.06
*ε_c_* _,*in*_	3.04 × 10^−3^	2.33 × 10^−3^	1.80 × 10^−3^	1.92 × 10^−3^
*c_i_* (μm)	0.1	0.1	0.1	0.1
*f_c_* _,*in*_	5.14%	4.06%	3.92%	4.21%
*X_in_*	0.075	0.090	0.060	0.093
*G_f_* (N/mm)	3.19 × 10^−7^	2.73 × 10^−7^	1.27 × 10^−6^	6.38 × 10^−7^

**Table 2 materials-16-05542-t002:** Chemical composition of the cement (wt%).

SiO_2_	Al_2_O_3_	Fe_2_O_3_	CaO	SO_3_	MgO	Na_2_O	K_2_O
19.95	4.63	2.95	61.87	2.51	2.09	0.15	0.66

**Table 3 materials-16-05542-t003:** Information about the experiments of ESA.

	Yu et al. [38]	Lee [45]	Cefis et al. [46]	Zhao et al. [44]	Xie et al. [47]
Material	Mortar	Mortar	Concrete	Concrete	Concrete
Cement type	PO 42.5	ASTM Type I	CEMII/A-LL	PC 32.5R	PO 42.5R
w/c	0.4	0.45	0.45	0.486	0.45
Corrosion solution	5% Na_2_SO_4_	5% Na_2_SO_4_	10% Na_2_SO_4_	10% Na_2_SO_4_	5% Na_2_SO_4_
Compressive test	Φ 50 × 100 mm	50 × 50 × 50 mm	Φ 150 × 300 mm	Φ 100 × 200 mm	Φ 150 × 300 mm
Expansion test	25 × 25 × 285 mm	25 × 25 × 285 mm	Φ 50 × 110 mm	Φ 100 × 200 mm	Φ 150 × 300 mm
Test period	Every month	91, 180, 270, 360, 450 d	Every two months until 1080 d	30, 90, 180, 270, 360 d	28, 91, 154, 217, 280 d

**Table 4 materials-16-05542-t004:** Model parameters for ESA simulation.

	Tests in This Study	Yu et al. [38]	Lee [45]	Cefis et al. [46]	Zhao et al. [44]	Xie et al. [47]
Material	Mortar	Mortar	Mortar	Concrete	Concrete	Concrete
*Y_in_* (GPa)	20.06	14.40	23.35	30.28	30.40	29.25
*Y_m_* (GPa)	24.62	17.85	29.58	34.62	36.50	40.78
*σ_c_*_,*in*_ (MPa)	34.37	29.18	47.60	30.02	30.00	25.00
*ε_c_* _,*in*_	2.75 × 10^−3^	2.33 × 10^−3^	3.81 × 10^−3^	1.66 × 10^−3^	1.66 × 10^−3^	1.47 × 10^−3^
*c_i_* (μm)	0.1	0.1	0.1	0.1	0.1	0.1
*γ*	0.04	0.04	0.04	0.025	0.025	0.025
*f_c_* _,*in*_	3.61%	4.06%	4.16%	2.67%	3.58%	3.11%
*X_in_*	0.082	0.090	0.081	0.075	0.094	0.098
*G_f_* (N/mm)	2.99 × 10^−7^	2.73 × 10^−7^	4.87 × 10^−7^	3.78 × 10^−7^	2.91 × 10^−7^	5.08 × 10^−7^

**Table 5 materials-16-05542-t005:** Fitting results of parameter *A*, *B*, and *C*.

	Tests in This Study	Yu et al. [38]	Lee [45]	Zhao et al. [44]	Xie et al. [47]
*A*	2.63 × 10^−2^	1.97	6.12 × 10^3^	2.44 × 10^−3^	6.08 × 10^3^
*B*	1.75 × 10^4^	44.4	2.32 × 10^−3^	6.52 × 10^−3^	7.70 × 10^−10^
*C*	4.93 × 10^9^	6.31 × 10^8^	2.12 × 10^8^	40.10	15.82

## Data Availability

The datasets generated during the current study are available from the corresponding author upon reasonable request.

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
