# Peer review of "A Micromechanical-Based Semi-Empirical Model for Predicting the Compressive Strength Degradation of Concrete under External Sulfate Attack"

_materials, 2023, doi:10.3390/ma16165542_

Round 1
Reviewer 1 Report
The work is relevant.
The paper presents the results of numerical modeling in comparison with experimental ones.
However, there are a few clarifying remarks:
1. In the simulation and in the experiment, sulfate is used as the effect of an aggressive environment. However, I ask you to emphasize the characteristics of sulfate by percentage effect. Was one value used, or as the sulfate ratio increased, how much did it affect the deformation?
2. Several works in this direction were presented in the study (Tables 1,4,5). At the same time, the difference with yours was not emphasized vividly. I ask you to explain what is the fundamental difference with these works, in addition to the fact that you have compared the simulation with experimental data.
3. The paper says that the difference between numerical experimental studies was about 10% under compression (Section 3). Here I ask you to emphasize at what ratio of sulfate exposure this difference was confirmed or sulfate exposure was not performed at this stage, but if an impact was made, then how it would affect the difference.
4. In conclusion, it is not clear what the authors propose as a scientific novelty. Results comparison of numerical and experimental studies or numerical research methodology. It is necessary to emphasize
No comments
Author Response
Thank you for your insightful comments and suggestions. Please see the attachment.

Reviewer 2 Report
The paper presents a study on predicting the compressive strength degradation of concrete under external sulfate attack (ESA). A micromechanical-based semi-empirical model is proposed, accounting for the amount of invading sulfate and chemical reactions. The model accurately describes the mechanical response of concrete under ESA, including compressive stress-strain behavior. Experimental validation confirms the relationship between compressive strength and expansion in sulfate-attacked material. The study predicts the deterioration process of compressive strength due to sulfate attack. The topic is interesting and falls within the aim of the journal. In addition, the results are well-presented and could be helpful to further develop the same topic. Therefore, the paper can be accepted for publication after considering following minor points:
- The main contribution of the research should be clearly discussed in the abstract. Please clearly state the novelty of the present study.
- Concerning the second-order equation governing the chemical reaction, are there any assumptions made that could affect the accuracy of the chemical degradation prediction?
- Are there any comparisons made between the proposed model and existing models for predicting ESA-induced compressive strength degradation? If so, how does the new model perform in comparison?
- Can it be applied to a wide range of concrete mixtures and environmental conditions, or are there specific limitations and conditions under which it may not be as accurate?
- Are there any potential implications for engineering practice and the design of concrete structures based on the findings of this study? Are there any recommendations for future research or areas for further investigation?
Author Response

(The authors gave the same response as above.)
